# Lorentzian contours for tree-level string amplitudes

**Lorenz Eberhardt,**[1] **Sebastian Mizera**[2]

[1]*Institute for Theoretical Physics, University of Amsterdam, Amsterdam, 1098XH, NL*

[2]*Institute for Advanced Study, Einstein Drive, Princeton, NJ 08540, USA*

*E-mail:* l.eberhardt@uva.nl, smizera@ias.edu

ABSTRACT: We engineer compact contours on the moduli spaces of genus-zero Riemann surfaces that achieve analytic continuation from Euclidean to Lorentzian worldsheets. These *generalized Pochhammer contours* are based on the combinatorics of associahedra and make the analytic properties of tree-level amplitudes entirely manifest for any number and type of external strings. We use them in practice to perform first numerical computations of open and closed string amplitudes directly in the physical kinematics for $n = 4, 5, 6, 7, 8, 9$. We provide a code that allows anyone to do such computations.

# Contents

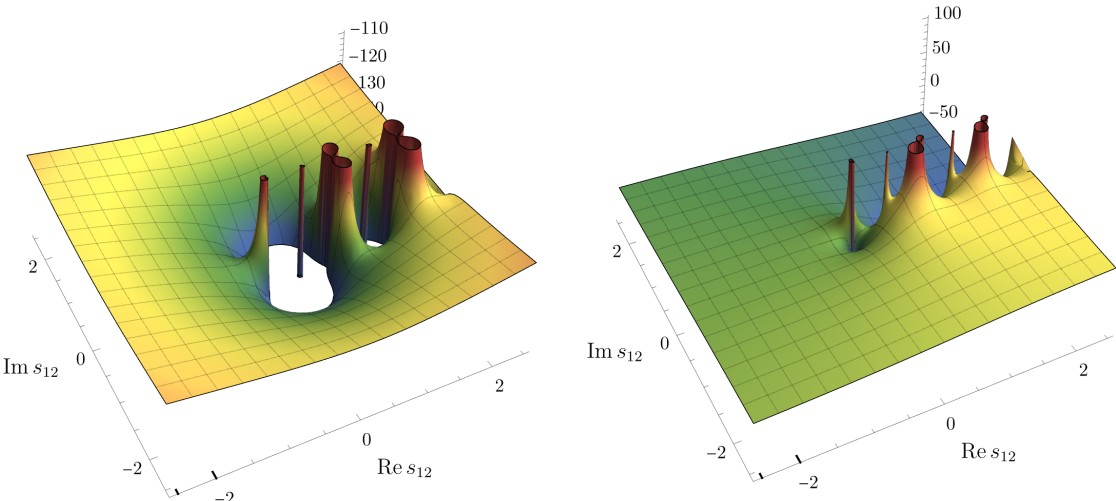

**Figure 1**. Example of a $2 \to 4$ tree-level open-string amplitude plotted in the complex plane of the total energy squared $s_{12}$. The spikes correspond to string resonances propagating in various channels. **Left:** Real part. **Right**: Imaginary part.

# 1 Introduction

Conventional description of string scattering as a computation on a Euclidean Riemann surface is in tension with the Lorentzian nature of space-time. For example, space-time unitarity and causal evolution remain obscured in this picture. At a more practical level, integration over all Euclidean worldsheet geometries (the moduli space of Riemann surfaces) leads to divergences. Indeed, this is precisely the reason why one cannot take a string amplitude appearing in a research paper and simply integrate it numerically. As pointed out in [1, 2], the Deligne–Mumford compactification does not fully address this problem and for good reason: the string integrand is not convergent regardless of whether it is integrated over the compactified or uncompactified moduli space, and even worse, starting at one loop the integrand does not even extend to a well-defined function over this compactification. A more careful and physical compactification is needed.

We studied this problem for four-point scattering amplitudes at genus one by proposing a Lorentzian contour of integration on the complexified moduli space in both open- and closed-string cases [1, 2]. This lead to a worldsheet understanding of unitarity cuts as well as new lessons about the physics of string interactions at one-loop level, including explicit computations of cross sections, partial waves, decay widths, high-energy limits, etc. However, generalizing this story to higher-point amplitudes requires a better understanding of the integration contour. This leads us to revisit the problem of computing higher-point amplitudes even at genus zero, which turns out to be a non-trivial problem.

This question has a mathematical and a physical angle. In 2002, Mimachi and

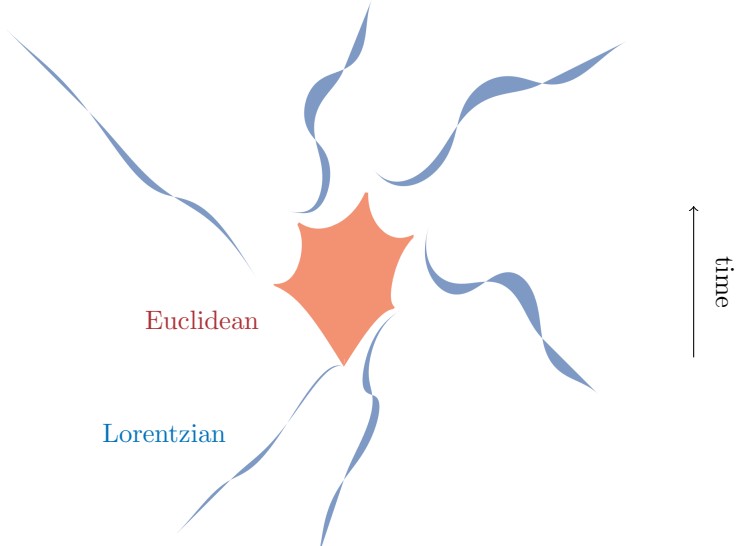

time

**Figure 2**. Example numerical plot of the worldsheet embedding in flat space for a $3 \to 3$ process (time goes up). The blue ribbon-like segments correspond to Euclidean evolution of strings, while the red region is where strings interact by tunneling into the Euclidean signature. For more details, we refer to [8, Sec. 7.3].

Yoshida described a compact integration contour $\Gamma_n$ for Selberg-like integrals which include genus-zero open string integrals [3, 4]. In 2013, Witten independently explained a physical prescription for designing a contour $\tilde{\Gamma}_n$ by Wick rotating from Euclidean to Lorentzian worldsheets near all its degenerations [5]. See Fig. 2 for an example of a worldsheet geometry resulting from such considerations. Effectively, $\Gamma_n$ can be thought of as a compact and resummed version of $\tilde{\Gamma}_n$, which are necessary properties for using it in the physical kinematics. We will refer to $\Gamma_n$ as the *generalized Pochhammer contour*. It was essential in applications of intersection theory, double-copy relations, and localization on boundaries of the moduli space [6, 7]. However, in those considerations only the *topology* of $\Gamma_n$ was important. Instead, the goal of this paper is to describe its *geometry* sufficiently accurately that it can be used in practical computations.

It is important that we learn how to address this problem directly in the physical (Lorentzian) kinematics. The reason is that studying string amplitudes in complex kinematics, even infinitesimally so, could lead to misleading conclusions due to Stokes phenomena at higher genus.

Our prescription for $\Gamma_n$ and its closed-string version $\Gamma_n^{\text{closed}}$ will make the analytic properties of genus-zero amplitudes completely manifest, i.e., all the resonance poles will appear as explicit prefactors. In fact, the same construction allows us to identify a much larger class of contours that define not only the amplitudes, but

also their generalized unitarity cuts. This geometry is based on the combinatorics of associahedra, which divide up the moduli space of punctured Riemann surfaces into regions associated with worldsheet configurations resembling individual Feynman diagrams (it is essentially the same combinatorics as in string field theory, see, e.g., [9]). An alternative approach to Lorentzian worldsheets could be that of Mandelstam's lightcone string diagrams [10, 11].

Numerical implementation of the contours described in this paper is given in the `Mathematica` notebook `LorentzianContours.nb` attached as an ancillary file to this submission. We refer directly to the notebook for the documentation of all the functions.

This paper is organized as follows. Sec. 2 reviews the topology of the generalized Pochhammer contours and their relation to Wick rotations and the combinatorics of associahedra. Sec. 3 uses this construction to give an explicit representation of these contours and shows how they can be used in practice. Sec. 4 repeats the analogous construction in the closed-string case. We finish in Sec. 5 with conclusions and outlook.

## 2 Topology of $\Gamma_n$

In this section, we review the construction of the generalized Pochhammer contour $\Gamma_n$ from a topological point of view. Most of the results in this section are previously known.

### 2.1 Setup

For concreteness, we are going to focus on $n$-point open-string amplitudes with planar ordering $123\cdots n$. Other orderings can be obtained by relabelling. Closed-string contour will be described in Sec. 4.

The general $n$-point disk amplitude in flat space can be written formally as

$$\mathcal{A}_n \overset{?}{=} \int\limits_{z_1 < z_2 < ... < z_n} \prod_{1 \leqslant i < j \leqslant n} (z_j - z_i)^{-\alpha' s_{ij}} \frac{\varphi(z_i)\,\mathrm{d}^n z}{\mathrm{SL}(2,\mathbb{C})}. \tag{2.1}$$

Here, $s_{ij} = (p_i + p_j)^2$ are the Mandelstam invariants of the external momenta $p_i$ and $\alpha'$ is the inverse string tension. The mass-squared $p_i^2$ is an integer multiple of $1/\alpha'$, i.e., vertex operators can be any massless, massive (or even tachyonic) states from the spectrum. From now on we set $\alpha' = 1$ for readability. The integrand has an $\mathrm{SL}(2,\mathbb{C})$ invariance, which allows us to fix positions of three vertex operators, say to $(z_1, z_{n-1}, z_n) = (0, 1, \infty)$. The integration then proceeds over the remaining positions $(z_2, z_3, \ldots, z_{n-2})$ such that their planar ordering is preserved. The rest of the integrand $\varphi(z_i)$ depends on the specific vertex operators that were used and hence

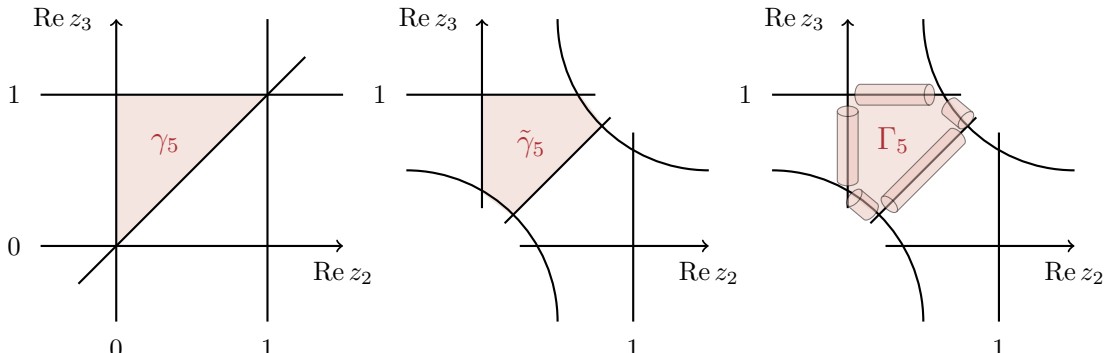

**Figure 3**. **Left:** The original contour $\gamma_5$ in the moduli space $\mathcal{M}_{0,5}$. Solid lines indicate points removed from the space at $z_2 = 0, 1$, $z_3 = 0, 1$, and $z_2 = z_3$. **Middle:** The same contour after compactification to $\widetilde{\mathcal{M}}_{0,5}$. New divisors appear as a result of blowing-up the points $(z_2, z_3) = (0, 0)$ and $(1, 1)$. **Right:** Cartoon of the generalized Pochhammer contour $\Gamma_5$. Each codimension-1 boundary of $\Gamma_5$ has a "tube" attached to it. See the main text for details.

can depend on the momenta, polarization vectors, etc. It is a rational function of $z_i$'s.

Only the multi-valued part of the integrand

$$(z_j - z_i)^{-\alpha' s_{ij}} \tag{2.2}$$

will be important for the discussion of the contour. We refer to it as the *Koba–Nielsen factor* [12]. It is also known that the general $n$-point function can be written as linear combinations of such integrals [13].

The question mark in (2.1) denotes the fact that the expression is only formal: it converges only in the region where all planar Mandelstam invariants are sufficiently negative, for which the integrand does not have any divergences. Hence one strategy would be to evaluate the integral in such a "safe" kinematic region and then analytically continue it to the physical region. Since $\mathcal{A}_n$ at tree-level are meromorphic functions of the Mandelstam invariants, such a continuation is unique. This approach, however, would require us to know the analytic form of $\mathcal{A}_n$. While for $n = 4$ and 5 such expressions are known [14], for $n \geqslant 6$ the amplitude cannot be represented in terms of the classic generalized hypergeometric functions $_pF_q$.

On the other hand, the approach of modifying the integration contour will allow us to evaluate $\mathcal{A}_n$ directly in the physical kinematics. This is the approach that has a hope of generalizing to higher-genus amplitudes as well.

## 2.2 Combinatorics

In order to describe the contour, we first need to understand the combinatorics and topology of the integration space. For the cloed string, this is the configuration space

of $n-3$ marked points on a Riemann sphere with 3 points removed. Let us call it $\mathcal{M}_{0,n}$. In coordinates, it can be written as

$$\mathcal{M}_{0,n} = \{(z_2, z_3, \ldots, z_{n-2}) \in \mathbb{C}^{n-3} \mid z_i \neq z_j \text{ for all } i \neq j \text{ from } i, j = 1, 2, \ldots, n\}.$$
(2.3)

In other words, we removed all configurations in which two punctures collide, which are therefore boundaries of the space. For this reason, $\mathcal{M}_{0,n}$ is non-compact.

The integration contour $\gamma_n = \{0 < z_2 < z_3 < \cdots < z_{n-2} < \infty\}$ used in (2.1) lies in the real subspace of $\mathcal{M}_{0,n}$. An example is given in Fig. 3 (left) for $n = 5$. It is a non-compact contour because it has endpoints at the boundaries of $\mathcal{M}_{0,n}$.

Note that boundaries of $\mathcal{M}_{0,n}$ also include more degenerate configurations in which three or more punctures collide. These can be blown-up with a Deligne–Mumford compactification $\widetilde{\mathcal{M}}_{0,n}$ of $\mathcal{M}_{0,n}$ [15]. It effectively allows us to resolve the rates at which particles collide, e.g., the triple degeneration $z_i = z_j = z_k$ can happen if $z_i$ collides with $z_j$ at a faster rate than $z_k$ with the $z_i = z_j$ system or vice versa.[1] As a result, $\gamma_n$ gets blown-up to $\tilde{\gamma}_n$. The new contour $\tilde{\gamma}_n$ is still non-compact. The Deligne–Mumford compactification does not cure divergences; it only exposes the combinatorics of worldsheet degenerations.

Compactifying each chamber (connected component) in the real part of $\mathcal{M}_{0,n}$ exposes its boundary structure. For example, for $n = 5$, the triangle $\gamma_5$ becomes a pentagon $\tilde{\gamma}_5$, see Fig. 3 (middle). For general $n$, the combinatorics of these chambers is described by the *Stasheff polytope* or the *associahedron*, $A_{n-1}$ [16]. It is an $(n-3)$-dimensional polytope. Its $k$-dimensional faces can be labelled by all ways of putting $n-k-2$ pairs of parentheses around $n-1$ ordered labels (by convention, we always put parenthesis around all the $n-1$ labels). For instance, $A_4$ has

$$k = 2: \quad (1234) \qquad\qquad\qquad\qquad\qquad\qquad\qquad\qquad \rightarrow 1 \text{ face}$$
$$k = 1: \quad ((12)34), \quad ((123)4), \quad (1(23)4), \quad (1(234)), \quad (12(34)) \quad\;\; \rightarrow 5 \text{ edges}$$
$$k = 0: \quad ((12)(34)), \quad (((12)3)4), \quad (1((23)4)), \quad ((1(23))4), \quad (1(2(34))) \;\; \rightarrow 5 \text{ vertices}$$

Two faces intersect if and only if their bracketings are compatible, i.e., they are either contained in one another or disjoint. Alternatively, each $k$-dimensional face can be described by a planar tree-level Feynman diagram, where each bracketing corresponds to a propagator. In other words, we can label each face by $n-k-3$ Mandelstam invariants $s_{I_1}, s_{I_2}, \ldots, s_{I_{n-k-3}}$ of a tree-level diagram. The real part of the moduli space is tiled by $(n-1)!/2$ such associahedra, which intersect at faces that are labelled by the same set of Mandelstam invariants. We refer to [6, 17] for more details on the combinatorics of associahedra and their relations to moduli spaces.

---

[1]As a simple example for why blow-ups are needed, consider expanding the function $\frac{1}{xy(x+y)}$ around the origin. One obtains different results depending how the origin is approached: taking $x \to 0$ first and then $y \to 0$ or vice versa.

## 2.3 Physical motivation

The next goal is to exploit the above combinatorics to describe the generalized Pochhammer contour $\Gamma_n$. We emphasize that $\Gamma_n$ is not a contour deformation of $\gamma_n$ or $\tilde{\gamma}_n$; contour deformation would of course not heal the endpoint divergences. Instead, $\Gamma_n$ is a new contour, depending on the Mandelstam invariants $s_{ij}$, such that for safe kinematics, for sufficiently negative $s_{ij}$, the two contours agree. While $\Gamma_n$ can be introduced purely mathematically through a procedure called *regularization* in twisted homology [18, Sec. 3.2] (see also [19] for a more complicated approach at $n = 5$), we will instead motivate it physically following [5] (see also [20] for previous work).

We will first focus on the simplest case, $n = 4$, for which

$$\mathcal{M}_{0,4} = \widetilde{\mathcal{M}}_{0,4} = \{z \in \mathbb{C} \mid z \neq 0, 1\}, \qquad \gamma_4 = \tilde{\gamma}_4 = (0, 1). \tag{2.4}$$

Here $z = \frac{(z_1 - z_2)(z_3 - z_4)}{(z_1 - z_3)(z_2 - z_4)} = z_2$ is the cross-ratio of the puncture coordinates. The Koba–Nielsen factor equals $z^{-s}(1 - z)^{-t}$, where $s = s_{12}$ and $t = s_{23}$. In the $s$-channel kinematics we have $s > 0$ and $t < 0$. Hence a possible divergence can come from the endpoint $z = 0$. The neighborhood of this point corresponds to geometries in which the worldsheet develops a long neck with some Schwinger proper time $\tau$. More concretely, it is related to the cross-ratio $z$ through $z = \mathrm{e}^{-\tau}$. Changing variables to $\tau$, the integrand behaves as

$$\mathcal{A}_4 = \int_0^\infty \mathrm{d}\tau\, \mathrm{e}^{\tau(s+k-1)}(c_0(t) + c_1(t)\mathrm{e}^{-\tau} + c_2(t)\mathrm{e}^{-2\tau} + \ldots). \tag{2.5}$$

where $k$ is the degree of the pole of $\varphi(z)$ at $z = 0$ and $c_i(t)$ are coefficients of the expansion of the integrand which in general depend on $t$. For example, $k = 1$ for scattering of gluons in superstring theory. Note that (2.5) still diverges as $\tau \to \infty$, but each term can be evaluated by analytic continuation to sufficiently negative $s$, giving

$$\mathcal{A}_4 = -\frac{c_0(t)}{s+k-1} - \frac{c_1(t)}{s+k-2} - \frac{c_2(t)}{s+k-3} - \ldots. \tag{2.6}$$

Hence, as expected, worldsheet degenerations in the $s$-channel are associated with a propagation of an infinite tower of states with masses-squared $1-k, 2-k, 3-k, \ldots$.

To obtain a better physical picture and avoid divergences, one has to analytically continue to Lorentzian worldsheets. This is strictly speaking only necessary for large $\tau > \tau_*$, where $\tau_* \gg 1$ is some arbitrary cutoff. This amounts to changing the contour according to

$$\int_0^\infty \to \int_0^{\tau_*} + \int_{\tau_*}^{\tau_* + i\infty}. \tag{2.7}$$

At this stage, there are two paths forward. One is to pretend that $s$ has a positive imaginary part, say $s + i\delta$ with $\delta > 0$. In this case, the integrand behaves as

$\mathrm{e}^{(\tau_* + iy)(s+i\delta)}$ and hence it is exponentially suppressed as $\mathrm{e}^{-y\delta}$ for large $y$ [5]. We will not take this approach because it has a serious drawback of breaking down as we approach the physical kinematics $\delta \to 0^+$. What one would observe in a numerical computation is that convergence would become poorer and poorer as we take $\delta \to 0^+$. Another problem is that the contour is still non-compact and hence will lead to divergences at resonances $s = 1-k, 2-k, 3-k, \ldots$, which are unavoidable in this approach.

As an alternative prescription, one can notice a quasi-periodicity of the integrand: as $\tau \to \tau + 2\pi i$, the integrand changes only by a phase $\mathrm{e}^{2\pi i s}$ (recalling that $k$ is an integer). Hence, (2.7) equivalently can be written as

$$\int_0^\infty \to \int_0^{\tau_*} + \int_{\tau_*}^{\tau_*+2\pi i} + \mathrm{e}^{2\pi i s} \int_{\tau_*}^{\tau_*+2\pi i} + \mathrm{e}^{4\pi i s} \int_{\tau_*}^{\tau_*+2\pi i} + \ldots \tag{2.8a}$$

$$= \int_0^{\tau_*} + \frac{1}{1 - \mathrm{e}^{2\pi i s}} \int_{\tau_*}^{\tau_*+2\pi i} . \tag{2.8b}$$

In the second line, we resummed the geometric series in $\mathrm{e}^{2\pi i s}$. Note that this procedure in itself requires a version of analytic continuation (say using $\mathrm{Im}\, s > 0$ for convergence), but it is done once and for all before any numerical computation starts. Back in the $z$ variable, the contour (2.8) amounts to replacing

$$\int_0^1 \to \frac{1}{1 - \mathrm{e}^{2\pi i s}} \int_{S_0^\varepsilon} + \int_\varepsilon^1 \tag{2.9}$$

where $0 < \varepsilon = \mathrm{e}^{-\tau_*} < 1$ and $S_0^\varepsilon$ denotes an anti-clockwise circle starting at $z = \varepsilon$ and centered at $z = 0$. We warn the reader that the first integral is not a residue because the integrand has a branch point at $z = 0$. The choice of a branch cut does not matter as long as the two pieces of contour are connected on the principal branch. In Fig. 4 we illustrate two equivalent choices in which the contour loops anti-clockwise and clockwise around $z = 0$, giving rise to different prefactors.[2]

So far, we have worked in the $s$-channel kinematics where $s > 0$ and $t < 0$. It was sufficient to consider the $s$-channel divergences as $z = 0$. In order to make the contour work in any kinematics, it suffices to repeat the same discussion around $z = 1$. This leads to the first example of the generalized Pochhammer contour:

$$\Gamma_4 = \frac{1}{1 - \mathrm{e}^{2\pi i s}} S_0^\varepsilon + (\varepsilon, 1 - \varepsilon) - \frac{1}{1 - \mathrm{e}^{2\pi i t}} S_1^{1-\varepsilon} , \tag{2.10}$$

where $S_1^{1-\varepsilon}$ starts at $z = 1 - \varepsilon$ and encircles $z = 1$ anti-clockwise. The extra minus sign arises because of the orientation of the contour. Here, $\varepsilon$ does not need to be infinitesimal, but only sufficiently small that it the circles do not enclose other

---

[2]The direction of the Lorentzian contour corresponds to the sign of the $i\varepsilon$ prescription in string theory. Hence it starts to matter for one-loop diagrams.

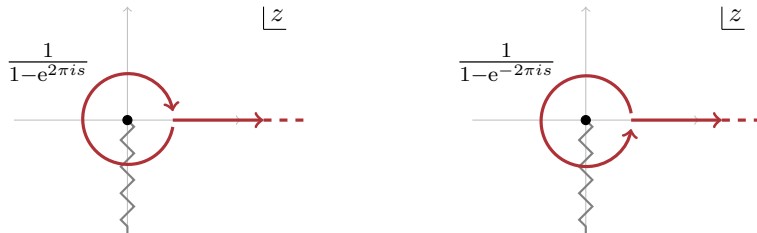

**Figure 4**. Two choices of orientations of the generalized Pochhammer contour near $z = 0$ differ by a sign in the prefactor. To go from the left to right contour we have to multiply by $e^{2\pi i s}$ because of the contour starts on a different branch and by $-1$ because of the orientation.

singularities, which means any value $0 < \varepsilon < 1$ will do. $\Gamma_4$ can be obtained by rearranging terms in the classic Pochhammer contour, which is why it is often referred to as the "Pochhammer contour" itself.

A few comments are in order. First of all, $\Gamma_4$ is compact and hence integrating over it cannot produce any divergences. All potential singularities of $\mathcal{A}_4$ are contained in the explicit prefactors in (2.10). They have an infinite number of poles when $s$ or $t$ is an integer. Of course, not all of them are actual singularities of $\mathcal{A}_4$ because their residue could vanish. For example, consider taking the residue around $s = m$:

$$\operatorname*{Res}_{s=m} \mathcal{A}_4 = \operatorname*{Res}_{s=m} \left[ \frac{1}{1 - e^{2\pi i s}} \oint_{S_0^\varepsilon} dz\, z^{-s-k}(c_0(t) + \ldots) \right] = -\operatorname*{Res}_{z=0} \left[ z^{-m-k}(c_0(t) + \ldots) \right].$$

(2.11)

The reason why now we could convert $S_0^\varepsilon$ into the residue is that at integer $s = m$, the integrand becomes single-valued around $z = 0$. We learn that the residue vanishes whenever $m < 1 - k$, which is consistent with the calculation above. For example, in superstring there are no poles for $m < 0$, i.e., no tachyons.

Secondly, if $s$ and $t$ are sufficiently negative such that the integrand is finite as $z \to 0$ and $z \to 1$, we can simply deform the loops $S_0^\varepsilon$ and $S_1^{1-\varepsilon}$ to points, i.e., take $\varepsilon \to 0^+$. In this case, $\Gamma_4 = \gamma_4$, which explains why the two contours are equivalent for safe kinematics.

## 2.4 Generalization to all $n$

The above discussion extends to $n > 4$, though we will see that it becomes difficult to realize $\Gamma_n$ concretely. Topologically, we can describe it as follows [4]. We start with the original contour $\tilde{\gamma}_n$ on the compactified moduli space, which is combinatorially an associahedron. Close to every codimension-1 degeneration, say at $D_I = 0$, the contour looks like an interval, say $(0, \mathcal{E})$, with an endpoint at $D_I = 0$ times an $(n - 4)$-dimensional contour. We simply replace this interval with a loop, just as in

:

$$\int_0^{\mathcal{E}} \to \frac{1}{1 - \mathrm{e}^{2\pi i s_I}} \int_{S_{D_I=0}^{\varepsilon}} + \int_{\varepsilon}^{\mathcal{E}}, \tag{2.12}$$

where $s_I$ is the Mandelstam invariant associated with a given degeneration $z_I = 0$. Hence every codimension-1 boundary of $\tilde{\gamma}_n$ is replaced with a "tube", see Fig. 3 (right) for a cartoon illustration. Close to every codimension-2 degeneration, one has to glue two tubes that intersect there and so on. Hence, locally close to every codimension-$k$ degeneration, the contour $\Gamma_n$ will look like a product of $k$ circles times $n - k - 3$ intervals, multiplied by the relevant kinematic factors.

The above discussion is formal because it does not tell us how to glue different pieces of the contour in practice. Describing this concretely will be the subject of Sec. 3.

Nevertheless, the above prescription is already powerful enough to study some aspects of string amplitudes which only depend on the topology of $\Gamma_n$ [7]. For example, as a generalization of the $n = 4$ discussion, it is clear that taking cuts of $\mathcal{A}_n$ localizes on the relevant divisors:

$$\operatorname*{Res}_{s_{I_1}=m_{I_1}} \cdots \operatorname*{Res}_{s_{I_k}=m_{I_k}} \mathcal{A}_n = (-1)^{n-3} \int_{\gamma_n \cap_{i=1}^{k} \{D_{I_i}=0\}} \operatorname*{Res}_{D_{I_1}=0} \cdots \operatorname*{Res}_{D_{I_k}=0} \tag{2.13}$$

$$\left[ \prod_{1 \leqslant i < j \leqslant n} (z_j - z_i)^{-\alpha' s_{ij}} \frac{\varphi(z_i)\, \mathrm{d}^n z}{\mathrm{SL}(2,\mathbb{C})} \right]_{i=1,2,\ldots,k}^{s_{I_i}=m_{I_i}}.$$

Here $I_1, I_2, \ldots, I_k$ is a set of $k \leqslant n-3$ compatible channels (for non-compatible ones the cut is zero). Hence cuts amount to a different choice of the integration contour on the moduli space. Likewise, the above construction also says that in the low-energy limit

$$\lim_{\alpha' \to 0} \Gamma_n = \bigcup_{\substack{\text{planar cubic} \\ \text{trees } T}} \prod_{I \in T} \frac{S_{D_I=0}^{\varepsilon}}{-2\pi i s_I}, \tag{2.14}$$

where the union is over all planar cubic trees $T$ and the product is over all subsets of labels $I$ defining such a tree. Therefore, if $\varphi$ is independent of $\alpha'$, then the $\alpha' \to 0$ limit of $\mathcal{A}_n$ is given by a sum of residues over maximal degenerations of the worldsheet. Masses of the particles or the behavior of $\varphi$ do not enter the discussion. In practice, it is useful to employ dihedral coordinates to preform such residues [21].

## 3 Geometry of $\Gamma_n$

The description of $\Gamma_n$ so far way implicit and not sufficient for practical integration. The goal of this section is to introduce the first explicit realization of this contour.

Following the discussion from the previous section, the contour will be decomposed into multiple pieces, one for each face of the associahedron, including the interior of the polytope. We can write

$$
\Gamma_n = \bigcup_{\substack{\text{faces} \\ (I_1, I_2, \ldots, I_k)}} \prod_{\ell=1}^{k} \frac{1}{1 - \mathrm{e}^{2\pi i s_{I_\ell}}} \times \Gamma_n^{(I_1, I_2, \ldots, I_k)}, \qquad (3.1)
$$

where each codimension-$k$ face is labelled by the set of $k$ compatible planar channels $(I_1, I_2, \ldots, I_k)$. We describe them in turn.

We first describe the Euclidean integration contour in suitable variables – the Wick rotation is then simple to perform in the end once we have set up everything correctly.

## 3.1  Interior

Let us first describe the interior $\Gamma_n^{(\varnothing)}$ of $\Gamma_n$. It will be convenient to write

$$
z_I = z_j - z_i \quad \text{where} \quad I = (i, \ldots, j). \qquad (3.2)
$$

Let us choose a set of constants $\varepsilon_I$, one for each facet of the associahedron. They are left as free parameters, up to some constraints described below, that can be used later on to improve numerical performance.

For sufficiently small $\varepsilon_I > 0$, imposing $z_I > \varepsilon_I$ for all planar channels $I$ carves out an associahedron:

$$
\Gamma_n^{(\varnothing)} = \{(z_2, z_3, \ldots, z_{n-2}) \in \mathbb{R}^{n-3} \mid z_I > \varepsilon_I \text{ for all planar } I\}. \qquad (3.3)
$$

This is an explicit realization of the associahedron without any blow-ups. For instance, the total number of inequalities that put constraints on the integration variables is $n(n-3)/2$. This is the same as the number of facets (codimension-1 faces) of the associahedron.

This construction tells us about some constraints we need to put on the constants $\varepsilon_I$. For the inequality $z_I > \varepsilon_I$ to produce a face of the associahedron, we must arrange that it is a genuinely new inequality not implied by any other one. This is straightforward, but somewhat tedious to work out since there are many inequalities to take into account. To simplify, we can assume that $\varepsilon_I$ only depends on the length of $I$, $\varepsilon_I \equiv \varepsilon_{|I|-1}$. The result is that we obtain an actual associahedron provided that the following inequalities are satisfied,

$$
2\varepsilon_{k-1} - \varepsilon_{k-2} < \varepsilon_k < \frac{1}{n-1-k}\big((n-2-k)\varepsilon_{k-1} + 1\big) \qquad (3.4)
$$

for $k = 1, \ldots, n-3$ and where $\varepsilon_k = 0$ for $k \leqslant 0$. We can pick any solution to these constraints. For concreteness, we can take the following solution for $\varepsilon_k$:

$$
\varepsilon_k = \frac{\varepsilon_1 k(k+1)}{2} \quad \text{with} \quad \varepsilon_1 < \frac{2}{n(n-3)}. \qquad (3.5)
$$

For the numerical implementation we chose $\varepsilon_1 = \frac{2}{n(n-1)}$, so that $\varepsilon_k = \frac{k(k+1)}{n(n-1)}$.

## 3.2 Vertices

Let us next consider the other extreme case with $k = n - 3$, i.e., when the face under consideration is a vertex. There are Catalan number $C_{n-2}$ such vertices. The Euclidean contour $\gamma_n^{(I_1,\ldots,I_{n-3})}$ (which as above is denoted by a lowercase $\gamma_n$) is given by displacing the original vertex along $n - 3$ Schwinger parameters

$$\widetilde{\Gamma}_n^{(I_1,\ldots,I_{n-3})} = \left\{ z_I = \varepsilon_I \, \mathrm{e}^{-\sum_{I_m \supseteq I} t_{I_m}} \text{ for } I \in \{I_1,\ldots,I_{n-3}\}, \; t_{I_m} \in [0,\infty) \right\}. \quad (3.6)$$

Here we introduced one Schwinger parameter $t_I$ for every face. In other words, for each bracket $I_i$, only the Schwinger parameters $t_m$ of its enclosing brackets $I_m \supseteq I_i$ appear for $m = 1, 2, \ldots, n - 3$.

For example, for the vertex $(((12)3)(45))$ we considered above, we have

$$(z_{123}, z_{12}, z_{45}) = (\varepsilon_2 \mathrm{e}^{-t_{123}}, \varepsilon_1 \mathrm{e}^{-t_{12}-t_{123}}, \varepsilon_1 \mathrm{e}^{-t_{45}}) \quad (3.7)$$

since $(I_1, I_2, I_3) = (123, 12, 45)$. Plugging this in into the Koba–Nielsen integrand brings it into the form

$$\int \mathrm{d}t_{123} \, \mathrm{d}t_{12} \, \mathrm{d}t_{45} \; \varepsilon_1^{2-s_{12}-s_{45}} \varepsilon_2^{1-s_{13}} \mathrm{e}^{(s_{45}-1)t_{45}+(s_{12}+s_{13}+s_{23}-2)t_{123}+(s_{12}-1)t_{12}} \left(1 - \mathrm{e}^{-t_{45}}\varepsilon_1\right)^{-s_{14}}$$

$$\times \left(1 - \mathrm{e}^{-t_{12}-t_{123}}\varepsilon_1\right)^{-s_{25}} \left(1 - \varepsilon_1 \mathrm{e}^{-t_{45}} - \varepsilon_1 \mathrm{e}^{-t_{12}-t_{123}}\right)^{-s_{24}} \left(\varepsilon_2 - \varepsilon_1 \mathrm{e}^{-t_{12}}\right)^{-s_{23}}$$

$$\times \left(1 - \varepsilon_2 \mathrm{e}^{-t_{123}}\right)^{-s_{35}} \left(1 - \mathrm{e}^{-t_{45}}\varepsilon_1 - \mathrm{e}^{-t_{123}}\varepsilon_2\right)^{-s_{34}}. \quad (3.8)$$

Our parametrization correctly implements the blowup of the moduli space near this vertex, as can be seen from the leading exponential dependence of the integrand on $t_I$. Notice in particular that the prefactor of $t_{123}$ in the exponent is $s_{123} - 2 = s_{12} + s_{13} + s_{23} - 2$, which leads to the correct poles of the amplitude.

## 3.3 Faces

We will now describe the parts of the contour associated to codimension-$k$ facets, which we denote by $\Gamma_n^{(I_1,I_2,\ldots,I_k)}$. This combines the two extreme cases that we considered above. To describe them systematically, we will first need to triangulate the whole associahedron. Even when we numerically integrate over the interior of the contour as defined in (3.3), the numerical integration strategy first triangulates the integration region.

In practice, we implemented the algorithm in [22], which gives a minimal triangulation of the associahedron in terms of $(n-2)^{n-4}$ simplices. In particular, we fix a triangulation that does not introduce new vertices. The triangulation of the interior of the associahedron then also induces a triangulation of the boundary facets (which

themselves are Cartesian products of lower-dimensional associahedra). Let us now consider one simplex on the codimension-$k$ facet.

On the codimension-$k$ facet $\Gamma_n^{(I_1,I_2,\ldots,I_k)}$, we can define the natural coordinates

$$z_{I_1}, \ z_{I_2}, \ \ldots, \ z_{I_k} . \tag{3.9}$$

However, in order to describe the $(n-3)$-dimensional contour, we need $n-k-3$ extra coordinates along the face. To define them, we remember that in a triangulation of say the interior of the contour (3.3), we can define coordinates $(\lambda_1, \ldots, \lambda_{n-3})$ in the interior of a vertex by

$$\sum_{\ell=1}^{n-2} \lambda_\ell \vec{z}_\ell , \tag{3.10}$$

where $\vec{z}_\ell$ are the locations of the vertices of the contour and $\lambda_{n-2} = 1 - \sum_{\ell=1}^{n-3} \lambda_\ell$ so that the first $n-3$ $\lambda_\ell$'s run over the region $\lambda_\ell \geqslant 0$ and $\sum_{\ell=1}^{n-3} \lambda_\ell \leqslant 1$.

Thus to define the remaining $n-k-3$ coordinates, we will linearly interpolate between the displaced vertices of a given simplex in the triangulation. Let $\vec{z}_1, \ldots, \vec{z}_{n-2+k}$ be the vertices of such a simplex. We then displace the vertices similarly as before. For a vertex defined by $(I_1, \ldots, I_k, I_{k+1}, \ldots, I_{n-3})$, we displace with the $k$ Schwinger parameters

$$z_I = \varepsilon_I e^{-\sum_{I_m \supseteq I, \, m \leqslant k} t_{I_m}} \text{ for } I \in \{I_1, \ldots, I_{n-3}\}, \ t_{I_m} \in [0, \infty) . \tag{3.11}$$

The only difference to (3.6) is that we have less Schwinger parameters. Let us denote the resulting positions of the vertices by $\vec{z}_\ell(t_{I_1}, \ldots, t_{I_k})$ for $1 \leqslant \ell \leqslant n - k - 2$.

When we interpolate between the vertices, we should not use the original $z$-coordinates, since this would not lead to the correct contour. Instead, we will define new coordinates $(y_1, \ldots, y_{n-3})$ (which depend on the simplex) and set

$$\widetilde{\Gamma}_n^{(I_1,\ldots,I_k)} = \bigcup_{\text{simplices}} \left\{ \vec{y} = \sum_{\ell=1}^{n-2-k} \lambda_\ell \vec{y}_\ell(t_{I_1}, \ldots, t_{I_k}), \ t_{I_m} \in [0, \infty), \ \lambda_\ell \geqslant 0, \ \sum_{\ell=1}^{n-2-k} \lambda_\ell = 1 \right\} . \tag{3.12}$$

This can then of course be translated back to the original coordinates.

Thus, the only remaining task is to define the coordinates $\vec{y}$. To define them, we need to choose a vertex at random. The definition of the coordinates $\vec{y}$ will depend on this choice, but this dependence can be absorbed into a redefinition of $\lambda_\ell$ and thus the actual contour is independent of this choice. Let $(I_1, \ldots, I_k, I_{k+1}, \ldots, I_{n-3})$ be the special vertex. For each $I_i$ from $i = 1, 2, \ldots, k$, we set $y_i = z_{I_i}$. For each $I_i$ from $i = k+1, \ldots, n$, we set $y_i = z_{I_i}/z_{\hat{I}_i}$. Overall, the new variables are

$$(y_1, y_2, \ldots, y_k, y_{k+1}, \ldots, y_{n-3}) = \left( z_{I_1}, z_{I_2}, \ldots, z_{I_k}, \frac{z_{I_{k+1}}}{z_{\hat{I}_{k+1}}}, \ldots, \frac{z_{I_{n-3}}}{z_{\hat{I}_{n-3}}} \right) . \tag{3.13}$$

From the definition (3.11), we see that the last $n-2-k$ coordinates are independent of the Schwinger parameters $t_{I_\ell}$, while the first $k$ coordinates are all identical for all vertices of the simplex. Because of this, it is then safe to interpolate between them as in (3.12). This property makes it also manifest that it does not matter which special vertex we choose in this definition, since on the level of the contour, it just amounts to a relabelling of the $\lambda_\ell$ parameters.

As an example, consider the codimension-2 facet $((12)3(45))$. It is a line with the two vertices $(((12)3)(45))$ (that we considered above) and $((12)(3(45)))$. Let us choose the first vertex to define the $y$-coordinates. Then the locations of these two vertices in $y$-coordinates takes the form

$$(y_1, y_2, y_3)(t_{12}, t_{45}) = (\varepsilon_1 e^{-t_{12}}, \varepsilon_1 e^{-t_{45}}, \varepsilon_2) , \tag{3.14a}$$

$$(y_1, y_2, y_3)(t_{12}, t_{45}) = (\varepsilon_1 e^{-t_{12}}, \varepsilon_1 e^{-t_{45}}, 1 - \varepsilon_2) . \tag{3.14b}$$

Clearly, it is trivial to interpolate between the two endpoints. When translating back to the original coordinates, we get the contour

$$z_2 = \varepsilon_1 e^{-t_{12}} , \qquad z_3 = 1 - \varepsilon_2 - \lambda_1(1 - 2\varepsilon_2) , \qquad z_4 = 1 - \varepsilon_1 e^{-t_{45}} . \tag{3.15}$$

By construction, this fits to the contour for the vertex that we constructed above.

## 3.4  Wick rotation

Once the contour is in the present form with good Schwinger parametrizations near all the facets, it is simple to Wick rotate. We can Wick rotate the Schwinger parameters $t_{I_m} \to i\, t_{I_m}$ that appear in a given facet $\Gamma_n^{(I_1,\ldots,I_k)}$. As explained above, we can restrict their range to $[0, 2\pi)$ and include the phase factor $(1 - e^{2\pi i s_{I_m}})^{-1}$, which gives the analytic continuation of the integral to arbitrary complex values of the Mandelstam invariants. We make two comments regarding the numerical implementation.

In practice, it is better to only Wick rotate the minimal subset of the Schwinger parameters that are necessary for convergence. To stay concrete, let us consider the case in which the integrand takes the form

$$\varphi(z_i) = \prod_{1 \leqslant i < j \leqslant n} (z_j - z_i)^{-n_{ij}} , \tag{3.16}$$

where $n_{ij}$ are integers. Conditions for convergence can then be expressed in terms of the shifted Mandelstam invariants $S_{ij} = \alpha' s_{ij} + n_{ij}$. Let us also use the convention

$$S_I := \sum_{\substack{i,j \in I \\ i<j}} (\alpha' s_{ij} + n_{ij}). \tag{3.17}$$

The original integral is already convergent near a face $I$ provided that

$$\mathrm{Re}(S_I) + 1 - |I| < 0 . \tag{3.18}$$

In these cases, there is a cancellation of poles in the Lorentzian contour (3.1), see the discussion around (2.11), which could lead to numerical instability. For such faces $I$, it is therefore computationally beneficial to use the original contour before Wick rotation.

In the numerical code, the criterion we choose to decide whether to Wick rotate around a given face $I$ is

$$\mathrm{Re}(S_I) + 1 - |I| > -\delta \, , \tag{3.19}$$

for some small positive parameter $\delta \in (0, 1)$. Setting $\delta$ too close to zero would have the effect of only barely regulating logarithmic divergences. Hence, in practice, we set $\delta = \frac{1}{2}$.

Second, one has to make sure in a numerical implementation that the integrand follows the choice of branch smoothly and does not jump when $t_{I_m} = \pi$. We do this in practice by isolating the biggest term in each basic factor $(z_i - z_j)^{-S_{ij}}$ and take it out of the parenthesis, which will lead to an exponential prefactor as in (3.8), which then automatically implements the correct branch.

## 3.5   Numerical examples

The attached `Mathematica` notebook implements the whole procedure. The function `A[s,ρ]` computes the open-string amplitude as a function of the Mandelstam invariant $s$ and the color ordering $\rho$ (by default $12\cdots n$). We refer directly to the notebook for the documentation of the options and conventions.

We have used the above implementation to evaluate string amplitudes up to $n \leqslant 9$ on randomly-chosen points in the kinematic space. Example plot obtained by computing the $n = 6$ case was shown in Fig. 1. The amplitude is plotted in the complex $s_{12}$ while keeping the other independent Mandelstam invariants fixed to $s_{13} = s_{14} = s_{23} = s_{24} = s_{25} = -s_{34} = -2s_{35} = -2s_{45} = -\frac{1}{4}$ and $n_{ij} = \delta_{i,i+1}$. The spikes in these plots correspond to resonances in the $s_{12}$, $s_{123}$, and $s_{56}$ channels. The numerical evaluation was made by requiring at least 3 digits of precision, which is enough for the purposes of a plot.

As another application, let us consider the fixed-angle high-energy limit of string scattering. This aspect is well-understood for $n = 4$, but little concrete results are available for $n > 4$. We are going to demonstrate how one can study this limit using numerical computations. For concreteness, we are going to study $2 \rightarrow n-2$ kinematics with $s_{ij} = \frac{1}{2}x\,\hat{p}_i \cdot \hat{p}_j$ with fixed directions $\hat{p}_i$ (corresponding to fixed scattering angles), as a function of $x$. We pick the incoming momenta $\hat{p}_1 = (1, 1, 0, 0)$ and $\hat{p}_2 = (1, -1, 0, 0)$ and symmetric configurations for the outgoing ones in which

$$n = 4 : \quad \hat{p}_3 = (-1, 0, 1, 0), \quad \hat{p}_4 = (-1, 0, -1, 0) \, , \tag{3.20a}$$

$$n = 5 : \quad \hat{p}_3 = (-\tfrac{2}{3}, 0, \tfrac{\sqrt{2}}{3}, -\tfrac{\sqrt{2}}{3}), \quad \hat{p}_4 = (-\tfrac{2}{3}, \tfrac{\sqrt{2}}{3}, -\tfrac{\sqrt{2}}{3}, 0), \tag{3.20b}$$

$$\hat{p}_5 = (-\tfrac{2}{3}, -\tfrac{\sqrt{2}}{3}, 0, \tfrac{\sqrt{2}}{3}) \, .$$

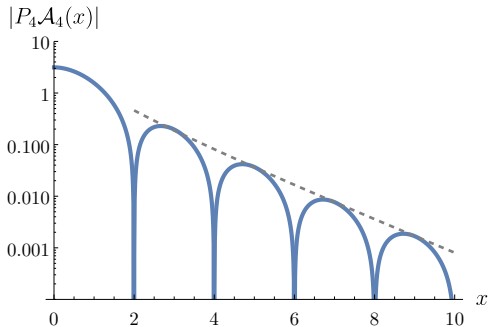
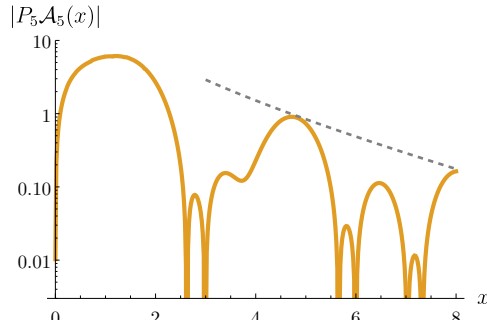

**Figure 5**. Plots of the normalized amplitude $|P_n\mathcal{A}_n(x)|$ for fixed-angle kinematics as a function of the energy parameter $x$. The plots are for $n = 4, 5$ respectively. The dashed lines indicate the asymptotics obtained from the MHV saddle point. Note the logarithmic scale.

Moreover, we use the integrands (3.16) with $n_{i,i+1} = 1$, which corresponds to Parke–Taylor forms [23].

In Fig. 5 we plot the corresponding absolute values of amplitudes on a logarithmic scale as a function of $x$, up to the prefactor $P_n$. The prefactor is simply there to remove poles at the positions of resonances so that the plots are easier to read. They are products of sines over all timelike planar Mandelstam invariants:

$$P_4 = \sin(\pi s), \quad P_5 = \sin(\pi s_{12})\sin(\pi s_{34})\sin(\pi s_{45}). \tag{3.21}$$

The downward spikes in Fig. 5 correspond to positions of zeros of the amplitudes. Plots for $n = 4, 5$ can of course be extended to arbitrarily large $x$ since they have known analytic expressions (implemented as A4 and A5 in the notebook). We note that these tree-level amplitudes admit a much milder and regular behavior compared to the one-loop corrections that are typically more erratic, see [2, Fig. 5].

The difficulty in analyzing this limit analytically is that there are $(n-3)!$ families of saddles, each one having an infinite number of images on different Riemann sheets of the Koba–Nielsen factor. This infinity is precisely what gives rise to the oscillating trigonometric functions multiplying exponential suppression $\sim e^{-S_n^{(a)}}$ for the $a$-th family of saddles. Depending on the value of the external kinematics, these different families are weighted with different functions $Q_n^{(a)}$, so that the overall behavior is

$$\mathcal{A}_n \sim \sum_{a=1}^{(n-3)!} Q_n^{(a)} e^{-S_n^{(a)}}. \tag{3.22}$$

Here, $S_n^{(a)}$ is the value of the string action evaluated on the $a$-th saddle. The functional form of $Q_n^{(a)}$ in terms of Mandelstam invariants and the scattering channel is not known beyond $n = 4$.

For four-dimensional kinematics, two saddles are always simple to describe (in the context of scattering equations, they are called the MHV and $\overline{\text{MHV}}$ solutions

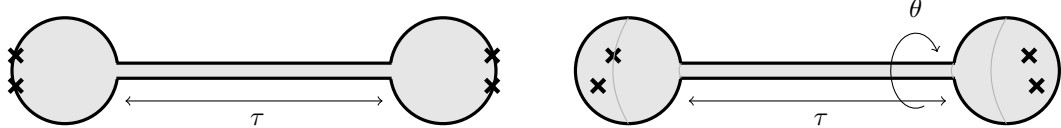

**Figure 6**. Example worldsheet geometries near which one modifies the integration contour from Euclidean to Lorentzian. The variable $\tau$ measures the Schwinger proper time along the long neck of the worldsheet and additionally $\theta$ measures the twist of the tube in the closed-string case.

[24, 25]). They are given by $z_i^{(1)} = \lambda_i^1/\lambda_i^2$ and $z_i^{(2)} = \tilde{\lambda}_i^{\dot{1}}/\tilde{\lambda}_i^{\dot{2}}$, where $(\lambda_i^\alpha, \tilde{\lambda}_i^{\dot{\alpha}})$ are the spinor-helicity variables of the $i$-th particle. Recall that for Lorentzian kinematics, we have $\tilde{\lambda}_i = \pm\lambda_i^*$, so the two families of solutions are complex conjugates of each other. It is straightforward to verify that $z_i^{(1/2)}$ satisfy the saddle point equations. In both cases, the exponential suppression comes from the real part of the on-shell action:

$$\operatorname{Re} S_n^{(1/2)} = \sum_{1 \leqslant i < j \leqslant n} s_{ij} \log |z_i^{(1/2)} - z_j^{(1/2)}| = \frac{1}{2} \sum_{1 \leqslant i < j \leqslant n} s_{ij} \log |s_{ij}| \,. \tag{3.23}$$

It is however not guaranteed that these saddles are dominant for $n > 5$.

In Fig. 5 we have overlaid the trend lines (dashed) corresponding to the exponential suppression of the MHV saddle point, $\sim \frac{a_n}{x^{(n-3)/2}} e^{-b_n x}$, where $b_n$ is the exponent obtained from (3.23) and $a_n$ is an arbitrarily-chosen constant that shifts the line up and down.

## 4 Closed-string contour

The goal of this section is to describe the analogous integration contour for closed-string amplitudes. Recall that they can be written as

$$\mathcal{A}_n^{\text{closed}} \overset{?}{=} \int_{\mathcal{M}_{0,n}} \prod_{1 \leqslant i < j \leqslant n} |z_j - z_i|^{-2\alpha' s_{ij}} \frac{\varphi_{\mathrm{L}}(z_i)\, \varphi_{\mathrm{R}}(\bar{z}_i)\, \mathrm{d}^n z}{\mathrm{SL}(2, \mathbb{C})} \,. \tag{4.1}$$

The first objective is to describe a physically-motivated compactification of $\mathcal{M}_{0,n}$, which can be viewed as a modified contour prescription in a larger space.

### 4.1 Wick rotation

Once again, it is enough to discuss Wick rotation locally close to the boundaries of the moduli space (which are real codimension 2 in this case). For this purpose, let us consider $n = 4$ in the $s$-channel kinematics where $s > 0$ and $t, u < 0$.

Consider the region $z \to 0$ where the singularity comes from. We can write the integrand in polar coordinates; the radius is then identified as $|z| = e^{-\tau}$, while we

have an additional angular coordinate and hence in total $z = e^{-t+i\theta}$. Here, $t$ can be thought of as the length of a long worldsheet neck (which becomes the Schwinger proper time in the field theory limit) and $\theta$ as parameterizing the twist of this neck (which does not have an obvious analogue in field theory), see Fig. 6.

As in (2.5), we can again expand the integrand in large $\tau$, which gives (for simplicity we set $\varphi_{\mathrm{L/R}} = 1$)

$$\mathcal{A}_4 = \int_0^\infty \mathrm{d}\tau \int_0^{2\pi} \mathrm{d}\theta \; e^{2\tau(s-1)}(c_{0,0}(t) + c_{1,-1}(t)e^{-\tau-i\theta} + c_{1,1}(t)e^{-\tau+i\theta} + \cdots) \,. \quad (4.2)$$

Besides the exponential corrections in the Schwinger parameter, we also get phases $e^{-\tau n+i\theta m}$ and coefficients $c_{n,m}$ with $-n \leqslant m \leqslant n$ and $n \equiv m \bmod 2$. The integral over $\theta$ implements the level-matching condition and keeps only the terms proportional to $c_{2n,0}$. One can then similarly formally compute the integral over $\tau$, which after analytic continuation leads to a series of poles. The physical Lorentzian contour is thus obtained similarly as above, by Wick rotating the Schwinger parameter $\tau$ to the upper half-plane after some large cutoff $\tau_*$.

## 4.2 Contour prescription

For practical computation, the open string contour was much nicer since it has a good combinatorial structure of the associahedron. For the closed string, no such structure exists since degenerations are real codimension 2 singularities. The generalization to arbitrary $n$ of the procedure sketched above can be achieved with a modification of the Kawai–Lewellen–Tye relations [26]. Recall that in their standard form, they treat the moduli space $\mathcal{M}_{0,n}$ as a contour embedded in $\mathcal{M}_{0,n}^{\mathrm{L}} \times \mathcal{M}_{0,n}^{\mathrm{R}}$, obtained by mapping all the left- and right-moving coordinates $(z_i, \bar{z}_i)$ into pairs of independent complex variables $(z_i, \tilde{z}_i)$.

The resulting expression takes the general form

$$\mathcal{M}_{0,n} = \sum_{\substack{\rho \in R \\ \tau \in T}} \gamma_\rho^{\mathrm{L}} \, S[\rho|\tau] \, \gamma_\tau^{\mathrm{R}} \,, \quad (4.3)$$

where the sum goes over two sets of orderings $\rho$ and $\tau$. The contours $\gamma_\rho^{\mathrm{L}} = \{z_{\rho(1)} < z_{\rho(2)} < \ldots < z_{\rho(n)}\}/\mathrm{SL}(2,\mathbb{R})$ and similarly for the $\gamma_\tau^{\mathrm{R}}$. We add superscripts $^{\mathrm{L/R}}$ to distinguish whether the contour is in $z_i$ or $\tilde{z}_i$ variables. Finally, $S[\rho|\tau]$ are meromorphic functions of the Mandelstam invariants that serve as the coefficients of the expansion.

Note that one has to specify the branch of the Koba–Nielsen factor on which $\gamma_\rho^{\mathrm{L}}$ and $\gamma_\tau^{\mathrm{R}}$ with different permutations are evaluated. We use the convention in which it equals $\prod_{\rho(i)<\rho(j)}(z_{\rho(j)} - z_{\rho(i)})^{-\alpha' s_{\rho(i)\rho(j)}}$ so that the integral over $\gamma_\rho^{\mathrm{L}}$ is real (away from poles) and likewise for $\gamma_\tau^{\mathrm{R}}$. In the language of twisted homology, this is associated with a canonical choice of *loading* a twisted cycle [18].

There are multiple equivalent ways to write (4.3). The most symmetric version takes all $(n-1)!/2$ inequivalent permutations $\rho$ and $\tau$ (up to cyclic rotation and reversal), for which $S[\rho|\tau]$ turn out to be phases that can read off from the combinatorics of the two permutations [27, Sec. 6.2]. For example, in the $n = 4$ case we have

$$\mathcal{M}_{0,4} = \tfrac{1}{2} \begin{bmatrix} \gamma^{\mathrm{L}}_{1234} \\ \gamma^{\mathrm{L}}_{1324} \\ \gamma^{\mathrm{L}}_{1342} \end{bmatrix}^{\mathsf{T}} \begin{bmatrix} 1 & \mathrm{e}^{-i\pi t} & \mathrm{e}^{i\pi s} \\ \mathrm{e}^{-i\pi t} & 1 & \mathrm{e}^{-i\pi u} \\ \mathrm{e}^{i\pi s} & \mathrm{e}^{-i\pi u} & 1 \end{bmatrix} \begin{bmatrix} \gamma^{\mathrm{R}}_{1234} \\ \gamma^{\mathrm{R}}_{1324} \\ \gamma^{\mathrm{R}}_{1342} \end{bmatrix} . \tag{4.4}$$

One the other hand, a more economical representation of (4.3) involves only $(n-3)!$ permutations of $\rho$ and $\tau$. In this case $S[\rho|\tau]$ are trigonometric functions of Mandelstam invariants, see [26], [28, App. A], or [29] for explicit expressions. These feature specific sets $R \neq T$ that minimize the number of total terms in the expansion (4.3), i.e., the number of zeros of the matrix $S[\rho|\tau]$.

For numerical purposes, one can actually make an even better choice by setting $R = T$. This is beneficial because the leading cost will be performing the integrals over the contours $\gamma^{\mathrm{L}}_{\rho}$ and $\gamma^{\mathrm{R}}_{\tau}$ and the above choice allows us to cut the computational time in half. For concreteness, we take

$$R = T = \{(1, \rho(2), \rho(3), \ldots, \rho(n-2), n-1, n) \text{ for } \rho \in S_{n-3}\}, \tag{4.5}$$

where $S_{n-3}$ is the set of permutations of $n-3$ labels. In this case, it is the simplest to use the representation of $S[\rho|\tau]$ as the inverse of the intersection matrix of the relevant contours $\mathbf{H} := [\gamma^{\mathrm{L}}_{\rho}|\gamma^{\mathrm{R}}_{\tau}]$:

$$S = \mathbf{H}^{-1}. \tag{4.6}$$

Even though for the choice $R = T$, this matrix is dense, $\mathbf{H}$ can be obtained efficiently using existing tools [30], and taking the inverse is subleading to the computation of the integrals over $\gamma^{\mathrm{L}}_{\rho}$. Overall, this becomes the most efficient strategy.

Finally, the discussion so far was formal because it involved non-compact cycles $\gamma^{\mathrm{L}}_{\rho}$ and $\gamma^{\mathrm{R}}_{\tau}$. As in the open-string case, we replace them by the generalized Pochhammer contours

$$\gamma^{\mathrm{L}}_{\rho} \to \Gamma^{\mathrm{L}}_{\rho} \qquad \gamma^{\mathrm{R}}_{\tau} \to \Gamma^{\mathrm{R}}_{\tau}. \tag{4.7}$$

Overall, the correct closed-string contour is given by replacing $\mathcal{M}_{0,n}$ with $\Gamma^{\mathrm{closed}}_{n} \subset \mathcal{M}^{\mathrm{L}}_{0,n} \times \mathcal{M}^{\mathrm{R}}_{0,n}$ defined according to

$$\boxed{\Gamma^{\mathrm{closed}}_{n} := \sum_{\substack{\rho \in R \\ \tau \in T}} \Gamma^{\mathrm{L}}_{\rho} \, S[\rho|\tau] \, \Gamma^{\mathrm{R}}_{\tau}.} \tag{4.8}$$

The previous discussion of unitarity cuts, low-energy limits, etc. can be easily generalized to the closed-string case by using this contour. In the next section we will also show that the analytic structure, with simple poles in all channels, is also made manifest.

As a simple example, consider $n = 4$. Here, we have

$$\Gamma_4^{\text{closed}} = \frac{\sin(\pi s)\sin(\pi t)}{\sin[\pi(s+t)]} \Gamma_{1234}^{\text{L}} \times \Gamma_{1234}^{\text{R}}. \tag{4.9}$$

Note the cancellation of poles: the product of the contours introduces a series of poles at integer $s$ and $t$, which are softened into simple poles by the sine factors in the numerator of the prefactor. Likewise, poles at integer $u$ are introduced by the denominator of this prefactor, since they are absent in each individual contour. These cancellations can be made manifest term-by-term since we already pulled out all divergences at the level of the generalized Pochhammer contours.

## 4.3 Worldsheet cuts

As explained in Sec. 2.4, the generalized Pochhammer contour allows us to construct also a number of simpler contours that compute unitarity cuts of the open-string amplitude. This can be done easily since all the analytic structure is made manifest by the prefactors in (3.1). Let us first summarize how the combinatorics works out for an arbitrary permutation $\rho$ of $n$ labels. Clearly, a unitarity cut in the $s_J$-channel is non-zero only if $J$ is compatible with the planar ordering $\rho$. The unitarity cut can then be computed as a residue around the given face $J$ of the associahedron, i.e., it only picks up contributions from a subset of terms in (3.1) of the form $\Gamma_n^{(\dots,J,\dots)}$. Moreover, the face itself can be written as a direct product of two smaller associahedra labelled by the permutations $(Jx)$ and $(\bar{J}\bar{x})$, where $\bar{J}$ is the complement of $J$ and $x$ (or $\bar{x}$ from the other side) is a new "emergent" puncture with momentum $p_x = -p_{\bar{x}} = -\sum_{j \in J} p_j$. Therefore, the resulting integration contour for the cut is given by

$$\underset{s_J = m_J}{\text{Res}} \Gamma_\rho = -\tfrac{1}{2\pi i}\left\{|D_J| = \varepsilon\right\} \times \Gamma_{(Jx)} \times \Gamma_{(\bar{J}\bar{x})}, \tag{4.10}$$

where the first piece represents the residue. Its orientation is induced by the orientation of $\Gamma_\rho$. Iterating this procedure (up to a maximum of $n-3$ times) gives unitarity cuts in multiple channels.

For example, starting with the planar ordering $(132456)$ and computing the cut in the labels $J = (132)$ leads to $(Jx) = (132x)$ and $(456x)$, i.e., the cut of the six-point amplitude factorizes into a product of two four-point amplitudes.

The generalization to closed strings is straightforward, see, e.g., [1, Sec. 2.2]. Locally close to the degeneration limit, the integration contour $\Gamma_n^{\text{closed}}$ takes the form of a direct product of the left- and right-moving coordinates. It is the simplest to work in the decomposition of $\Gamma_n^{\text{closed}}$ that passes through the relevant degeneration at $(D_J, \tilde{D}_J) = (0,0)$. It takes the form of a double-contour integral in the $(D_J, \tilde{D}_J)$ planes, illustrated in Fig. 7, times the remaining $(n-4)$-dimensional component. After unifying the orientations as in Fig. 4, each of the left- and right-moving contours has the coefficient $\frac{1}{1-e^{-2\pi i s_J}}$, but the kernel behaves as $\frac{1}{2i}(1-e^{-2\pi i s_J})$. Overall, it leads

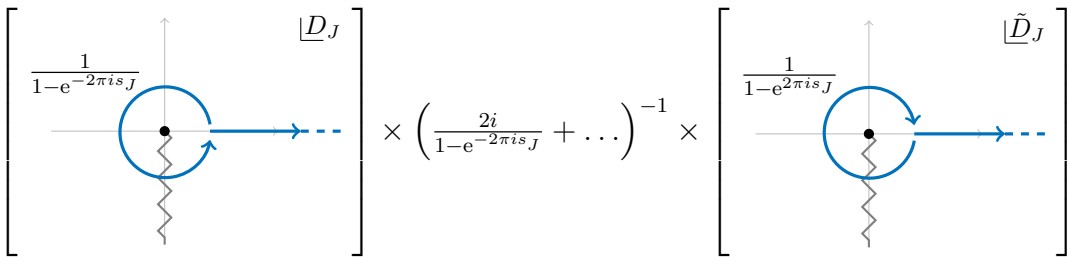

**Figure 7**. Part of the integration contour $\Gamma_n^{\text{closed}}$ intersecting the $(D_J, \tilde{D}_J)$ planes close to the degeneration at the origin. The kinematic prefactors manifest the singularity structure and allow one to cleanly compute unitarity cuts.

to at most simple poles at integer values of $s_J$. The integration contour therefore once again manifests the singularity structure of closed string amplitudes.

Unitarity cuts are now simple to compute by picking up the residue at $s_J = m_J$:

$$\operatorname*{Res}_{s_J = m_J} \Gamma_n^{\text{closed}} = -\tfrac{1}{4\pi} \{|D_J| = \varepsilon\} \times \{|\tilde{D}_J| = \varepsilon\} \times \Gamma_{|J|+1}^{\text{closed}} \times \Gamma_{|\tilde{J}|+1}^{\text{closed}}, \tag{4.11}$$

where, as before, the last two terms correspond to the contours of the factorized amplitudes to the left and the right of the cut. In particular, the value of the cut is non-zero only if both residues are simultaneously non-zero. Multiple cuts are obtained by iterating the same procedure in non-overlapping channels up to the maximum of $n - 4$ times.

As an example, consider the Virasoro–Shapiro amplitude for $n = 4$:

$$\mathcal{A}_4^{\text{closed}} = \int_{\Gamma_4^{\text{closed}}} \mathrm{d}^2 z \, (z\tilde{z})^{-s-1} \left[(1-z)(1-\tilde{z})\right]^{-t-1}. \tag{4.12}$$

Let us apply the above prescription to the $u$-channel cut (where $u = -s - t$), which originates from $z \to \infty$. Following the above contour prescription, we have

$$\operatorname*{Res}_{u=m} \mathcal{A}_4^{\text{closed}} = -\frac{1}{4\pi} \oint_{z, \tilde{z}=\infty} \mathrm{d}^2 z \, (z\tilde{z})^{-s-1} [(1-z)(1-\tilde{z})]^{m+s-1} \tag{4.13}$$

for integer $m$. At infinity, the integrand behaves as $\sim (z\tilde{z})^{m-2}$, which means that it only has poles when $u = m \in \mathbb{Z}_{\geqslant 1}$. For example, when $m = 0, 1, 2$ we get

$$\operatorname*{Res}_{u=0} \mathcal{A}_4^{\text{closed}}(s,t) = \pi \left[\operatorname*{Res}_{z=\infty}[0 + \mathcal{O}(\tfrac{1}{z^2})]\right]^2 = 0, \tag{4.14a}$$

$$\operatorname*{Res}_{u=1} \mathcal{A}_4^{\text{closed}}(s,t) = \pi \left[\operatorname*{Res}_{z=\infty}[\tfrac{1}{z} + \mathcal{O}(\tfrac{1}{z^2})]\right]^2 = \pi, \tag{4.14b}$$

$$\operatorname*{Res}_{u=2} \mathcal{A}_4^{\text{closed}}(s,t) = \pi \left[\operatorname*{Res}_{z=\infty}[1 - \tfrac{s+1}{z} + \mathcal{O}(\tfrac{1}{z^2})]\right]^2 = \pi(s+1)^2. \tag{4.14c}$$

One can verify these results using the explicit form of $\mathcal{A}_4^{\text{closed}}$ in terms of Gamma functions.

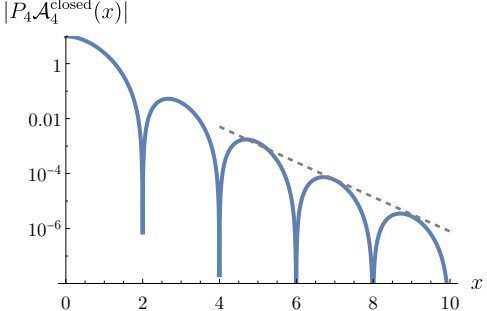 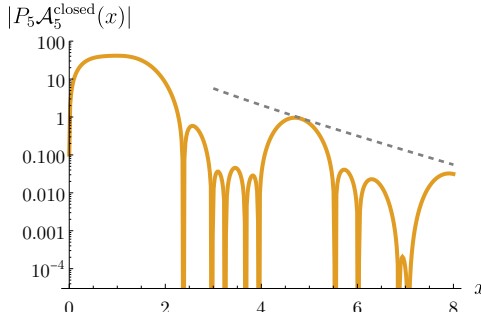

**Figure 8**. Numerical plots of the absolute value of the closed-string amplitude normalized to $|P_n\mathcal{A}_n^{\text{closed}}(x)|$. It is the closed-string counterpart of Fig. 5.

## 4.4 Numerical results

We have implemented a numerical code for closed-string computations in the attached `Mathematica` notebook. The function `Aclosed[s]` does this computation as a function of the Mandelstam invariants, see the notebook for documentation.

As a simple application, we repeated the analysis of Sec. 3.5 for closed strings. We use exactly the same kinematics for $n = 4, 5$. The results are shown in Fig. 8, together with the trend lines showing the asymptotics of the MHV saddle point whose slope is twice as steep as in the open-string case in Fig. 5.

## 5 Conclusion

In this work, we constructed compact integration contours for genus-zero open and closed string amplitudes implementing their Lorentzian time evolution. They are given in (3.1) and (4.8) and implemented in the `Mathematica` notebook attached to this submission. Let us mention a few interesting points and future directions.

**Shift relations.** For completeness, we should mention that there is another approach to compute the amplitudes at tree-level in physical kinematics. We did not pursue it since it does not generalize well to higher-genus amplitudes. It is based on the observation that the amplitude satisfies various shift relations that come from integration by parts identities. For example, for

$$\mathcal{A}_4(s, t) = \int_0^1 \mathrm{d}z \ z^{-s} \, (1 - z)^{-t} \ , \tag{5.1}$$

the following identities hold:

$$\mathcal{A}_4(s + 1, t) = \frac{s + t - 1}{s} \, \mathcal{A}_4(s, t) \ , \qquad \mathcal{A}_4(s, t + 1) = \frac{s + t - 1}{t} \, \mathcal{A}_4(s, t) \ . \tag{5.2}$$

Repeated application of these shift relations lets one bring the kinematics to sufficiently negative Mandelstam variables for which the moduli space integral over the

Euclidean contour converges. This approach generalizes to higher-point amplitudes, where one can act with shift operators $s_{ij} \rightarrow s_{ij} + 1$ on a vector of $(n-3)!$ amplitudes. The result is the same vector multiplied by a *contiguity matrix*, see, e.g., [31, Sec. 4.1]. Repeated use of such relations can also bring one to the safe region of Mandelstam invariants where integrals can be evaluated numerically on the original contour.

We anticipate that the implementation of this approach at genus zero would actually be *more* efficient than the numerical implementation of the complicated contour that we pursued in this paper. We were however reluctant to follow this path, since it is not physically well-motivated and, to our knowledge, does not generalize to higher-loop amplitudes. Moreover, for genus-zero planar amplitudes, anomaly cancellation requires one to sum over the planar annulus and Möbius strip contributions. Combined, they have branch cuts when $s \geqslant 0$, $t \geqslant 0$, and $u \geqslant 0$, which means the safe Euclidean region of kinematics does not exist.

**String field theory and Mandelstam lightcone prescription.** A related prescription inspired by string field theory for the computation of amplitudes as discussed in this paper was proposed in [32]. It is again difficult to generalize to higher loops and also more difficult to implement practically even at tree level, since it involves solving differential equations. An alternative approach could be Mandelstam's lightcone diagrams, which treats string worldsheets as Lorentzian from the get go. Some historical references in applying them to loop computations include [20, 33, 34].

**Open-closed scattering.** We discussed scattering of open or closed strings separately, but one can of course also consider mixed open-closed scattering amplitudes. In some cases, there are extensions of the KLT formula available that also express mixed amplitudes fully in terms of open string scattering amplitudes (at least with one closed string insertion) and thus one could straightforwardly extend the techniques discussed in this paper to that case [35, 36]. However, one gets a linear combination of higher-point open string amplitudes with collinear external momenta in this case. This could increase the complexity of the numerical evaluation.

**Higher loops.** Arguably the most interesting and pressing problem is to develop an extensive package for perturbative string calculations at one- and possibly two-loops. In this case, one needs to systematically implement the integration contour similarly to what we have done here. The contour does in general no longer have the form of a polyhedron, but there are still combinatorial descriptions available [37, 38]. We expect that such a package would be indispensable if one wants to further explore the physical properties of string amplitudes.

**Compact contours for Feynman integrals.** Finally, let us mention a possibility of applying the techniques developed in this paper to numerical evaluation of

Feynman integrals. In dimensional regularization, multi-loop Feynman integrals are structurally almost identical to tree-level string amplitudes. For example, the combinatorics of graph degenerations is captured by "Feynman polytopes" [39, 40] which are the counterparts of the associahedra. A compact integration contour based on their combinatorics would allow one to manifest the singularity structure in the dimensional regulator $\varepsilon$ and hence allow for a direct numerical integration of infrared and ultraviolet-divergent integrals, perhaps utilizing the $u$-variables from [41]. It would be interesting to understand whether such an approach can be made practical.

## Acknowledgements

We thank Carolina Figueiredo for discussions. S.M. gratefully acknowledges funding provided by the Sivian Fund and the Roger Dashen Member Fund at the Institute for Advanced Study. This material is based upon work supported by the U.S. Department of Energy, Office of Science, Office of High Energy Physics under Award Number DE-SC0009988.

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
