# Peer review of "Lorentzian contours for tree-level string amplitudes"

_SciPost Physics_

## Round 1 · Referee Report · Anonymous (Referee 1) · 2024-4-16

Report

The manuscript under review presents integration contours in multiparticle string tree-level amplitudes which cure the divergences of the underlying moduli-space integrals in the physical kinematic region and make numerical evaluations accessible. The results of the paper close a long-standing gap in the literature and line up with a broader, successful research agenda of the authors to manifest unitarity properties of string amplitudes and to enable actual ``evaluations''. Related studies of integration contours in one-loop string amplitudes in the authors’ earlier work proved extremely valuable to harness the poor convergence properties of the relevant four-point moduli-space integrals at genus one. The achievements of the present paper at genus zero are indispensable for a systematic approach to controling analyting properties of multi-loop and -leg string amplitudes and making numerical evaluations possible.

The manuscript is very well-written and convinces through a concise presentation while providing the necessary background information. For instance, a sophisticated combinatorial structure arising from scattering an arbitrary numbers of external legs is explained in a transparent way. Several figures and plots neatly illustrate important points in the main text. A thoroughly documented mathematica notebook is provided as an ancillary file which demonstrates the practical use of the papers’ results and will be really helpful to colleagues working on related problems.

I can only offer minor suggestions for improvements in the final version and happily recommend the manuscript for publication in SciPost once the authors have considered the points below.

Requested changes

1) It’s a delight to read about the implications for fixed-angle high-energy limit of string amplitudes in section 3.5. I agree with the authors that "little concrete results are available for $n > 4$" points in this limit. It could further strengthen the manuscript if the authors already give a pointer to the high-energy discussion in the introduction, make it more visible by adjusting the subsection headline(s) and slightly expand section 4.4 by comments on the high-energy limit in the closed-string case.

2) When the authors mention Stokes phenomena at higher genus in the introduction, it would be useful to give a reference at the end of that paragraph.

3) The Moebius transformations relevant for the open string form an $SL(2,\mathbb R)$, not an $SL(2,\mathbb C)$. Please adjust this in (2.1), in the fifth line below, in (2.13) and potentially other places (below (4.3), the contour $\gamma^{\rm L}_\rho$ is correctly quotiented by $SL(2,\mathbb R)$).

4) Even though one example is implicit in (4.9), it would not hurt to spell out one or two instances of the $S[ \rho | \tau ]$ matrix (and possibly ${\bf H}$) in the authors’ basis choice (4.5) at some point in section 4.2.

5) In the paragraph on ``Open-closed scattering’’ in the conclusion, also 0907.2211 should be cited along with references [35, 36].

6) Please fix the typos on the word "way" (the 1st line of section 3) and "cloed string’’ (2nd line of section 2.2).

Recommendation

Ask for minor revision

---

## Round 1 · Referee Report · Anonymous (Referee 2) · 2024-8-7

Report

In this article the authors address the question on how to perform the integration over the moduli space required for tree-level string amplitudes with Lorentzian kinematics. Requiring Lorentzian kinematics makes the problem non-trivial.

The authors first study open-string amplitudes. Starting from the moduli space ${\mathcal M}_{0,n}$ and its compactification $\widetilde{\mathcal M}_{0,n}$ they carefully construct an appropriate integration contour $\Gamma$ and provide computer routines for the actual integration. The method is clearly explained. I appreciate the combination of first giving an algorithm for the solution of the problem together with examples of computer implementations.

For the closed-string amplitudes they use KLT-relations to bring the problem back to open-string amplitudes.

Overall this is a very nice paper, which I am happy to recommend for publication.

A few small comments to improve the presentation:

  1. In section the authors start with the open-string amplitudes and they explicitly say that closed-string amplitudes will be treated in section 4, but in section 2.2 all of a sudden the moduli space for the closed-string amplitudes appears, before the real sub-space relevant to the open-string amplitudes is introduced. Maybe rewording the introduction of section 2.2 slightly could help, such that readers are not left with the impression that some information is missing here.

  2. Page 6 below: "cloed" --> "closed"

Recommendation

Publish (easily meets expectations and criteria for this Journal; among top 50%)

---

## Editorial Decision

resubmitted